# Sex and prior exposure jointly shape innate immune responses to a live herpesvirus vaccine

**Foo Cheung**[1†], **Richard Apps**[1†], **Lesia Dropulic**[2#], **Yuri Kotliarov**[1], **Jinguo Chen**[1], **Tristan Jordan**[2¶], **Marc Langweiler**[1], **Julian Candia**[1], **Angelique Biancotto**[1], **Kyu Lee Han**[1], **Nicholas Rachmaninoff**[3], **Harlan Pietz**[2**], **Kening Wang**[2], **John S Tsang**[1,3*‡††], **Jeffrey I Cohen**[2*‡]

[1]Center for Human Immunology, National Institutes of Health, Bethesda, United States; [2]Medical Virology Section, Laboratory of Infectious Diseases, National Institutes of Health, Bethesda, United States; [3]Multiscale Systems Biology Section, Laboratory of Immune System Biology, National Institutes of Health, Bethesda, United States

*For correspondence: john.tsang@yale.edu (JST); jcohen@niaid.nih.gov (JIC)

†These authors contributed equally to this work
‡These authors also contributed equally to this work

Present address: #Vaccine Research Center, National Institutes of Health, Bethesda, United States; ¶George Washington University School of Medicine and Health Sciences, Washington, DC, United States; **Weill Cornell Medicine and The Rockefeller University, New York, United States; ††Department of Immunobiology and Center for Systems and Engineering Immunology, Yale University School of Medicine; Department of Biomedical Engineering, Yale University, New Haven, United States

**Competing interest:** The authors declare that no competing interests exist.

## Abstract

**Background:** Both sex and prior exposure to pathogens are known to influence responses to immune challenges, but their combined effects are not well established in humans, particularly in early innate responses critical for shaping subsequent outcomes.

**Methods:** We employed systems immunology approaches to study responses to a replication-defective, herpes simplex virus (HSV) 2 vaccine in men and women either naive or previously exposed to HSV.

**Results:** Blood transcriptomic and cell population profiling showed substantial changes on day 1 after vaccination, but the responses depended on sex and whether the vaccinee was naive or previously exposed to HSV. The magnitude of early transcriptional responses was greatest in HSV naive women where type I interferon (IFN) signatures were prominent and associated negatively with vaccine-induced neutralizing antibody titers, suggesting that a strong early antiviral response reduced the uptake of this replication-defective virus vaccine. While HSV seronegative vaccine recipients had upregulation of gene sets in type I IFN (IFN-α/β) responses, HSV2 seropositive vaccine recipients tended to have responses focused more on type II IFN (IFN-γ) genes.

**Conclusions:** These results together show that prior exposure and sex interact to shape early innate responses that then impact subsequent adaptive immune phenotypes.

**Funding:** Intramural Research Program of the NIH, the National Institute of Allergy and Infectious Diseases, and other institutes supporting the Trans-NIH Center for Human Immunology, Autoimmunity, and Inflammation. The vaccine trial was supported through a clinical trial agreement between the National Institute of Allergy and Infectious Diseases and Sanofi Pasteur. Clinical trial number: NCT01915212.

## Editor's evaluation

This important study uses a systems immunology approach to disentangle the effect of sex and prior exposure on an individual's response to viral vaccines. This work brings an impressive data assortment and present compelling evidence to support the conclusions. The paper would be of interest to researchers working in human immunology and vaccine development.

## Introduction

Systems vaccinology aims to improve understanding of vaccine outcomes, by using unbiased approaches to identify the major associated immune phenotypes, including peripheral blood cell population frequencies and gene expression (*Pulendran et al., 2010*). This has been particularly informative for dissecting the molecular and cellular correlates of natural human variation that can influence vaccine responses and outcomes in vivo (*Tsang, 2015*). Sex and prior exposure to the same or similar antigen are prominent examples of natural variation that influence vaccinee responses. However, immune response differences associated with sex and prior exposure have typically been defined and analyzed separately. These variables have not been examined together because vaccination studies often involve individuals who are either almost entirely previously exposed (e.g., influenza studies *Bucasas et al., 2011*; *Nakaya et al., 2016*; *Nakaya et al., 2011*; *Tsang et al., 2014*; *Kotliarov et al., 2020*) or naive (e.g., Ebola or yellow fever studies in US cohorts *Gaucher et al., 2008*; *Querec et al., 2009*; *Rechtien et al., 2017*) to the vaccine pathogen. Here, we utilize a unique cohort of herpes simplex virus (HSV) 2 vaccine recipients that includes both sexes, as well as naive and previously exposed subjects, thus allowing us to study the in vivo molecular and cellular response correlates of the joint effects of sex and prior exposure.

Sex is a well-known variable that can influence vaccine and disease responses. Females typically develop higher antibody responses and report more adverse reactions following vaccination than males (*Klein et al., 2015*). These differences could be a result of multiple sex dimorphic traits, including sex hormones and increased expression of X-linked genes escaping inactivation. Females in particular have been observed to show higher levels of Toll-like receptor (TLR)/interferon (IFN)-associated gene expression in innate responses to vaccination (*Klein et al., 2010*). Systems immunology studies have identified prevaccination differences that predict antibody titer responses, such as CD20$^+$CD38$^{high}$ B cell frequency and plasmacytoid dendritic cell activation (*Kotliarov et al., 2020*). Prior exposure to antigen is another dominant factor in immune response, but its effects have most often been analyzed focusing on adaptive, antigen-specific responses. An example of particular interest for innate responses involved vaccine vectors using replication-defective viruses: trials testing an experimental adenovirus vector-based HIV vaccine revealed increased inflammatory responses in subjects seropositive for a prevalent adenovirus subtype and showed an enhanced risk of HIV infection that was associated with prior adenovirus exposure (*Zak et al., 2012*; *Buchbinder et al., 2008*; *Gray et al., 2011*).

Here, we analyzed a unique cohort receiving a multidose, replication-defective vaccine, to interrogate the interaction of sex and prior exposure to infection. Subjects received an experimental HSV2 vaccine in a phase 1 clinical trial where comparisons could be made between three groups of volunteers based on their HSV serostatus prior to vaccination: HSV1$^-$/HSV2$^-$, HSV1$^+$/HSV2$^-$, or HSV1$^\pm$/HSV2$^+$ (*Dropulic et al., 2019*). Each of these groups included subjects from both sexes. HSV2 is an important human pathogen. Persons infected with HSV2 have a threefold increased risk of acquiring HIV, and HSV2 infection causes genital herpes and severe disease in neonates, patients with AIDS, and transplant recipients (*Koelle and Corey, 2008*; *Freeman et al., 2006*). Prior attempts to produce an effective prophylactic vaccine for HSV2, including glycoprotein subunit or replication-competent vaccines, have been unsuccessful (*Dropulic and Cohen, 2012*). The replication-defective vaccine HSV529 studied here is deleted for two essential HSV genes, UL5 and UL29, and propagated in a cell line expressing these two proteins (*Da Costa et al., 1999*; *Da Costa et al., 2000*). Infection of cells not expressing these two essential genes results in expression of nearly all the viral genes, but viral DNA replication is abolished, and no virions are produced. The vaccine induced greater than fourfold increases in HSV-specific antibody titers in most HSV1$^-$/HSV2$^-$ vaccine recipients, but lower increases in antibody responses were observed in volunteers in the other serogroups; T cell responses were modest in all three serogroups (*Dropulic et al., 2019*).

We characterized immune responses longitudinally by analyzing blood transcriptomic profiles and high-dimensional immune cell phenotyping throughout the three-dose vaccine regimen (*Dropulic et al., 2019*). Vaccine-induced changes and correlates of outcomes such as neutralizing antibody titers and adverse reactions to the vaccine were determined. The transcriptomic responses were associated with both sex and prior exposure to HSV, but the responding genes and pathways also differed markedly based on the combination of both variables. Women showed stronger early inflammatory and type I IFN responses when compared to men, but this was only observed for HSV seronegative vaccine recipients. Thus, using unbiased systems immunology analyses, we show how

prior exposure and sex interact to shape early, innate immune responses to a replication-defective vaccine, and identify transcriptional profiles that may shape the subsequent immune responses induced in vivo.

## Materials and methods

### Vaccine recipients and study design

All HSV529 vaccine recipients signed informed consent for a protocol (clinicaltrials.gov ID: NCT01915212) approved by the National Institute of Allergy and Infectious Diseases Institutional Review Board (*Dropulic et al., 2019*). The HSV529 vaccine was manufactured by Sanofi Pasteur. Peripheral blood was obtained at various timepoints and PBMCs were isolated by separation with Ficoll-Paque PLUS and cryopreserved. A web tool for visualization of experimental designs was used to summarize the samples and assays available at sampled timepoints (*Cheung and CHI Consortium, 2020*).

### RNA sequencing

Fresh PBMCs were collected and lysed in TRIzol (Thermo Fisher, Waltham, MA), and total RNA was isolated and purified with an miRNeasy kit (Qiagen, Hilden, Germany). All blood samples at different timepoints from the same subject were processed together. RNA quality and quantity were estimated using Nanodrop (Thermo Scientific, Wilmington, DE) and Agilent 2100 Bioanalyzer (Agilent Technologies, Palo Alto, CA). Before sequencing analysis, all samples were batched according to their age, sex, race, and immunization status, but assayed blindly. Two reference samples were simultaneously processed with the vaccine study samples in each batch. Stranded cDNA sequencing libraries were generated with a TruSeq Stranded mRNA Library Prep Kit (Illumina, San Diego, CA) following the manufacturer's instructions. Briefly, 500 ng of total RNA was used for mRNA selection. After the reverse transcription to first strand cDNA, strand information was maintained with dUTP during second strand synthesis. A single nucleotide (containing adenine) was added to the dsDNA fragments and the products were ligated to an adapter. The products were then purified and amplified by PCR to create the final cDNA library. The library was qualified with an Agilent Bioanalyzer and quantified with a Qubit 2.0 fluorometer. The cluster generation and pair-end (2 × 75 bp) sequencing were performed on an Illumina HiSeq 3000. Up to 96 barcoded samples were pooled for one single run, which yielded at least 30 M passed filter paired reads per sample. Sequencing results were demultiplexed and converted to FASTQ format using Illumina bcl2fastq software. The sequencing reads were adapter and quality trimmed and then aligned to the human genome using STAR software, and read counts determined with HTSeqCount software. Raw read counts were normalized using the DESeq2, LIMMA packages, and R software.

### Cells and flow cytometry

Viable PBMCs were isolated and cryopreserved according to Center for Human Immunology (CHI) protocols (https://chi.niaid.nih.gov/web/new/our-research/sop.html). High parameter flow cytometry was performed using the Human Immunology Project Consortium (HIPC) panels as previously described (*Maecker et al., 2012*; *Finak et al., 2016*; *Langweiler and McCoy, 2019*). Briefly, 4 parallel 10-color panels with a total of 26 unique markers enabled detection of 70 subsets of PBMCs represented as a fraction of their parent population (*Supplementary file 1A, B*).

Staining was performed using dedicated lyophilized antibody plates for each of compensation, fluorescence-minus-one controls, and study sample staining (all BD Biosciences). Sample staining plates included up to 10 study samples in addition to control PBMCs from a healthy donor, and staining was preceded by an additional incubation with LIVE/DEAD Fixable Blue Dead Cell Stain (Thermo Fisher Scientific). Acquisition was performed with a Becton Dickinson LSRFortessa, using DIVA 8 software, acquiring 250,000 cells for each sample. Subsequent analysis to determine population frequencies used FlowJo version 9.6.2. Compensation performed with unstained cells and Becton Dickinson compensation beads was used to aid acquisition monitoring. Subsequently a final compensation matrix was calculated using FlowJo during postacquisition analysis.

## Statistics and computational analysis

Batching of samples, quality control filtering, and statistical analyses were performed using R/Bioconductor as well as R-Shiny web tools similar to those previously published (*Cheung et al., 2017*; *Cheung, 2023,* copy archived at swh:1:rev:5480d6351a4740b56297f2470b24402bd1b676b9). Longitudinal changes in flow cytometry populations were evaluated by Wilcoxon's signed-rank paired test, and p values were corrected for multiple comparisons using the Benjamini–Hochberg method, using an R webtool. For transcriptomic responses the most significantly responding genes were initially identified using Bioconductor package DESeq2 and then responses were further analyzed using linear models for microarray data (LIMMA) (*Smyth, 2004*; *Love et al., 2014*). Low expressed genes were removed and day 1 expression after subtraction of baseline values were used to fit models based on log counts per million, with an interaction term included when comparing the effect of sex between groups based on prior exposure. The *tmod* package in R was used for blood transcription module (BTM) analysis (*Zyla et al., 2019*). Each BTM is a set of genes which has been shown to show coherent expression across many biological samples in conditions including in vivo responses to interventions such as vaccines (*Bar-Joseph et al., 2003*; *Chaussabel and Baldwin, 2014*). BTM analysis can used to identify significant enrichment of a set of foreground genes, in predefined transcriptional modules compared against a reference set. The hypergeometric test devised in *tmodHGtest* was used to calculate enrichments and p values employing Benjamini–Hochberg correction for multiple sampling. All the statistical analyses and graphical presentations were performed in R. Gene set variation analysis (GSVA) was used to quantify subject-level variation in signatures of interest (*Zyla et al., 2019*; *Hänzelmann et al., 2013*).

## Reactions to HSV529 vaccination

Solicited systemic and local reactions to vaccine were graded based on a toxicity table as reported previously (*Dropulic et al., 2019*). Severity of disease, represented by aggregate symptom scores, wase determined by multiplying the severity of each symptom by the number of days it persisted. Welch's unpaired *t*-test was used to determine statistically significant differences (two-tailed $p < 0.05$) in aggregate symptom scores between women and men after each vaccine dose.

# Results

## Vaccination strategy and study design

Sixty subjects who comprised three groups based on their HSV serostatus prior to vaccination were studied: HSV1$^-$/HSV2$^-$, HSV1$^+$/HSV2$^-$, and HSV1$^\pm$/HSV2$^+$ (*Dropulic et al., 2019*). The three groups each consisted of 20 volunteers who received intramuscular injection with either HSV529 vaccine (15 participants) or placebo (5 participants), and 50% of the vaccine recipients were women. The mean age of vaccine recipients was 31 years (range 21–40) and did not differ significantly between the groups defined by serology and sex. The vaccine regimen consisted of three vaccinations and peripheral blood samples were obtained on the day before each of these vaccinations, and at follow-up timepoints for all 60 subjects (*Figure 1A*). Bulk RNA-seq was performed at nine timepoints and flow cytometry at seven timepoints including the day of each vaccination, 1 day after the first vaccination, and 7 days after each of the three vaccinations. Measurements were also obtained for neutralizing antibody titers to HSV, complete blood count tests, and adverse reactions to the vaccination at various timepoints (*Figure 1A*).

## Prior exposure and sex are associated with the kinetics of the antibody response and adverse events induced by vaccination

The HSV529 vaccine induced increases in HSV-specific antibody titers in most recipients (*Dropulic et al., 2019*). These responses were most prominent in HSV1$^-$/HSV2$^-$ vaccine recipients, while in subjects previously exposed to HSV the vaccine responses were more modest particularly at later timepoints. Kinetic analysis of the profiles of HSV2 neutralizing antibody titers showed clear differences based on prior exposure to HSV. In subjects who were HSV1$^-$/HSV2$^-$ before vaccination, neutralizing antibody responses peaked at 1 month after the third dose of vaccine at day 210 (Wilcoxon's signed-rank paired test, $p < 0.05$), and antibody titers declined significantly by day 360 regardless of sex ($p < 0.001$) (*Figure 1B*). In contrast, individuals seropositive for HSV1 prior to vaccination

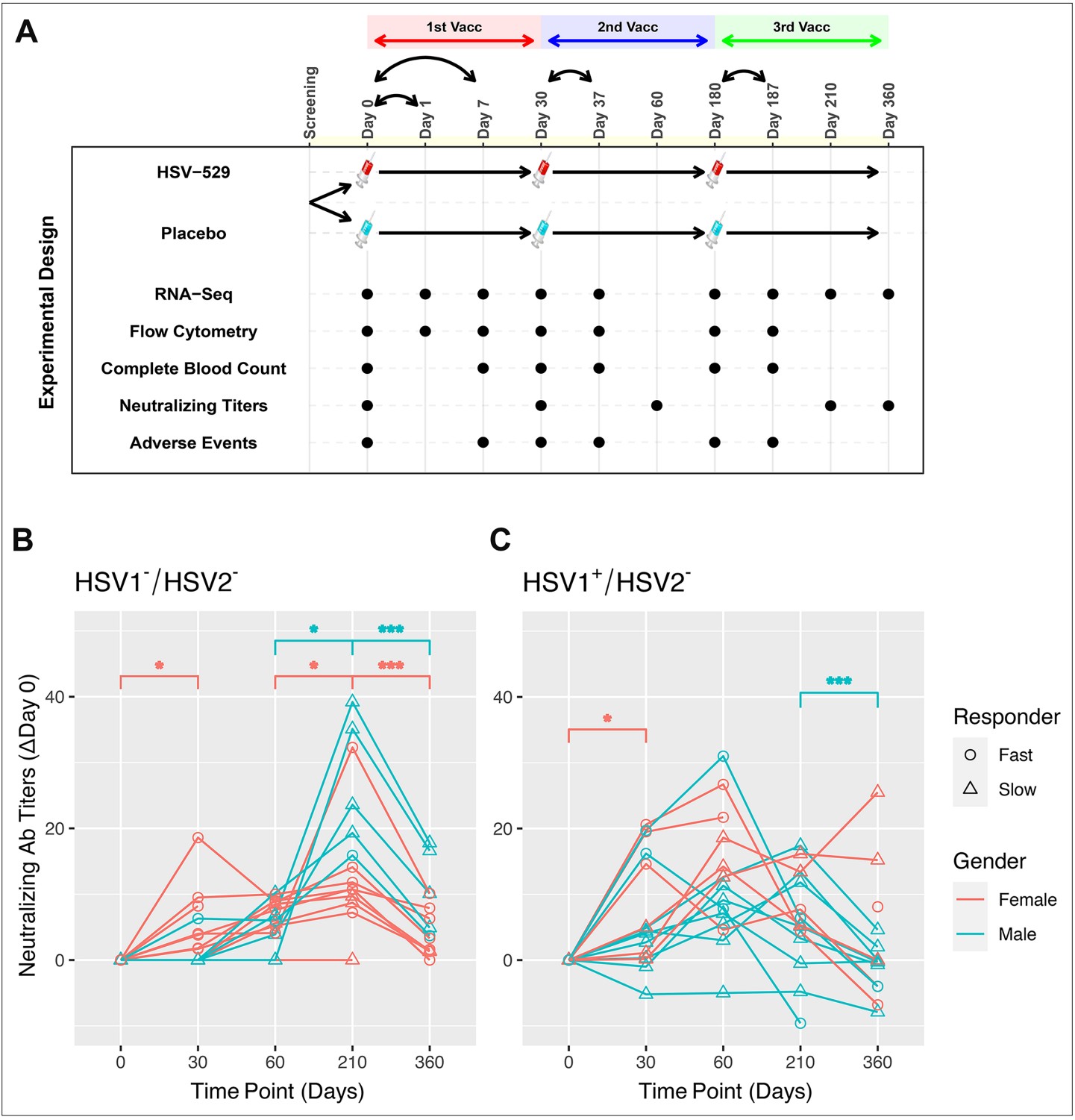

**Figure 1.** Prior exposure and sex affect neutralizing antibody responses after HSV529 vaccination. (**A**) Schematic outline of the vaccination strategy and study design. Each subject was randomized to receive three vaccinations with HSV529 or saline placebo on days 0, 30, and 180. Timepoints are marked at which blood was obtained for immune phenotyping assays, or when adverse events were scored. (**B, C**) Changes in HSV2 neutralizing antibody titer over time are shown for HSV529 vaccine recipients, plotted separately for subjects with no prior exposure to herpes simplex virus (HSV) (HSV1⁻/HSV2⁻) or for HSV1 seropositive subjects (HSV1⁺/HSV2⁻). Subjects are shown classified by sex and as slow or fast responders to vaccination based on the increase in HSV2 neutralizing titer by day 30. Rapid responders were defined by an increase in neutralizing antibody responses at day 30 for subjects who were HSV1⁻/HSV2⁻ before vaccination. In subjects HSV1⁺/HSV2⁻ prior to vaccination, in whom an increase in neutralizing antibody responses was

*Figure 1 continued on next page*

*Figure 1 continued*

observed by day 30 for most individuals, rapid responders were a distinct group marked by higher titer responses compared to slow responders at day 30. Significant changes are indicated for men and women separately (*p < 0.05, ***p < 0.001).

The online version of this article includes the following figure supplement(s) for figure 1:

**Figure supplement 1.** Quantification of adverse events in HSV529 recipients.

showed neutralizing antibody responses that peaked at day 60 or earlier for the majority of subjects (*Figure 1C*). Neutralizing antibodies were only tested at day 0 and 210 in HSV2⁺ vaccine recipients, so for this group kinetics of the response could not be assessed.

Effects of sex were observed particularly when focusing on kinetics of the neutralizing antibody responses to the first vaccination dose. Neutralizing antibodies increased significantly by day 30 compared to baseline in HSV1⁻/HSV2⁻ and HSV1⁺/HSV2⁻ women, but not men; and this increase was most significant for HSV1⁻/HSV2⁻ women (p = 0.022) (*Figure 1B, C*). Within both the HSV1⁻/HSV2⁻ and HSV1⁺/HSV2⁻ groups, subjects could be further separated into groups of approximately equal sizes of rapid and slow responders based on the neutralizing antibody responses observed by day 30 (*Figure 1B, C*). Women were more highly represented than men in the fast responder groups, with this most pronounced for HSV1⁻/HSV2⁻ subjects in which women comprised 89% of fast responders (*Figure 1B*).

Evaluation of adverse events observed in this cohort also indicated effects of both serostatus and sex. HSV1⁻/HSV2⁻ women reported more systemic and local reactions to the vaccine than men. About 65% of HSV529 recipients had systemic reactions to the vaccine (compared with ~55% of placebo recipients), while ~90% of HSV529 recipients had local injection site reactions (compared with ~50% of placebo recipients) (*Dropulic et al., 2019*). Analysis of sex-specific differences in HSV1⁻/HSV2⁻ vaccine recipients showed that women were more likely to have more severe systemic reactions within the first 7 days of the first dose of vaccine than men (Welch's unpaired *t*-test, p = 0.015) and more severe local reactions within the first 7 days of the second dose of vaccine (p = 0.017) (*Figure 1— figure supplement 1*). However, differences in systemic or local reactions were not noted between women and men in the HSV1⁺/HSV2⁻ or HSV1±/HSV2⁺ groups. Together, these results demonstrate that both sex and prior exposure to HSV1 markedly affect the response to this vaccine with HSV1⁻/HSV2⁻ women showing the most robust early responses marked by a more rapid induction of HSV2 neutralizing antibody, and a higher frequency of adverse events.

## The magnitude of the response to vaccination is greatest at day one after the first vaccine dose

For all 60 subjects, peripheral blood bulk RNA-seq and high parameter flow cytometry measurements were analyzed from seven matched timepoints, and at two further timepoints for RNA-seq (*Figure 1A*). Initially all subjects that received HSV529 were combined to identify genes differentially expressed longitudinally in response to vaccination, and gene set enrichment analysis was performed using BTMs to identify the highest responding biological pathways (*Figure 2A, B*; *Subramanian et al., 2005*; *Li et al., 2014*). Many of the responses previously observed in other vaccine studies were detected (*Nakaya et al., 2011*; *Tsang et al., 2014*; *Querec et al., 2009*; *Kazmin et al., 2017*). Inflammatory and IFN pathways, dendritic cell, T cell, and B cell responses were all detected at day 1 after the first vaccination. Cell cycling, T cell, and B cell responses were observed at day 7 after the first vaccination. A timepoint of 7 days after vaccination was chosen for analysis of the subsequent doses, and although after the first vaccine dose the magnitude of changes was reduced at the individual gene level, in terms of gene set enrichments the changes for subsequent doses resembled those after the first dose, particularly for dose 3. The individual genes most significantly upregulated on day 1 after the first dose of vaccine included those encoding IFN-induced proteins and Fc receptors (*Supplementary file 1C*).

Multi-parameter flow cytometry quantified 70 cell populations which were compared for the same timepoints as the transcriptomic analysis, for all subjects combined. Cell population changes occurring across broad lineages were observed as expected for vaccine responses (*Figure 2C* and *Supplementary file 1D*). Monocyte and NK cell frequencies significantly increased at day 1 after the first dose of vaccine, whereas the Th1 subset decreased, possibly reflecting polarization of the vaccine response.

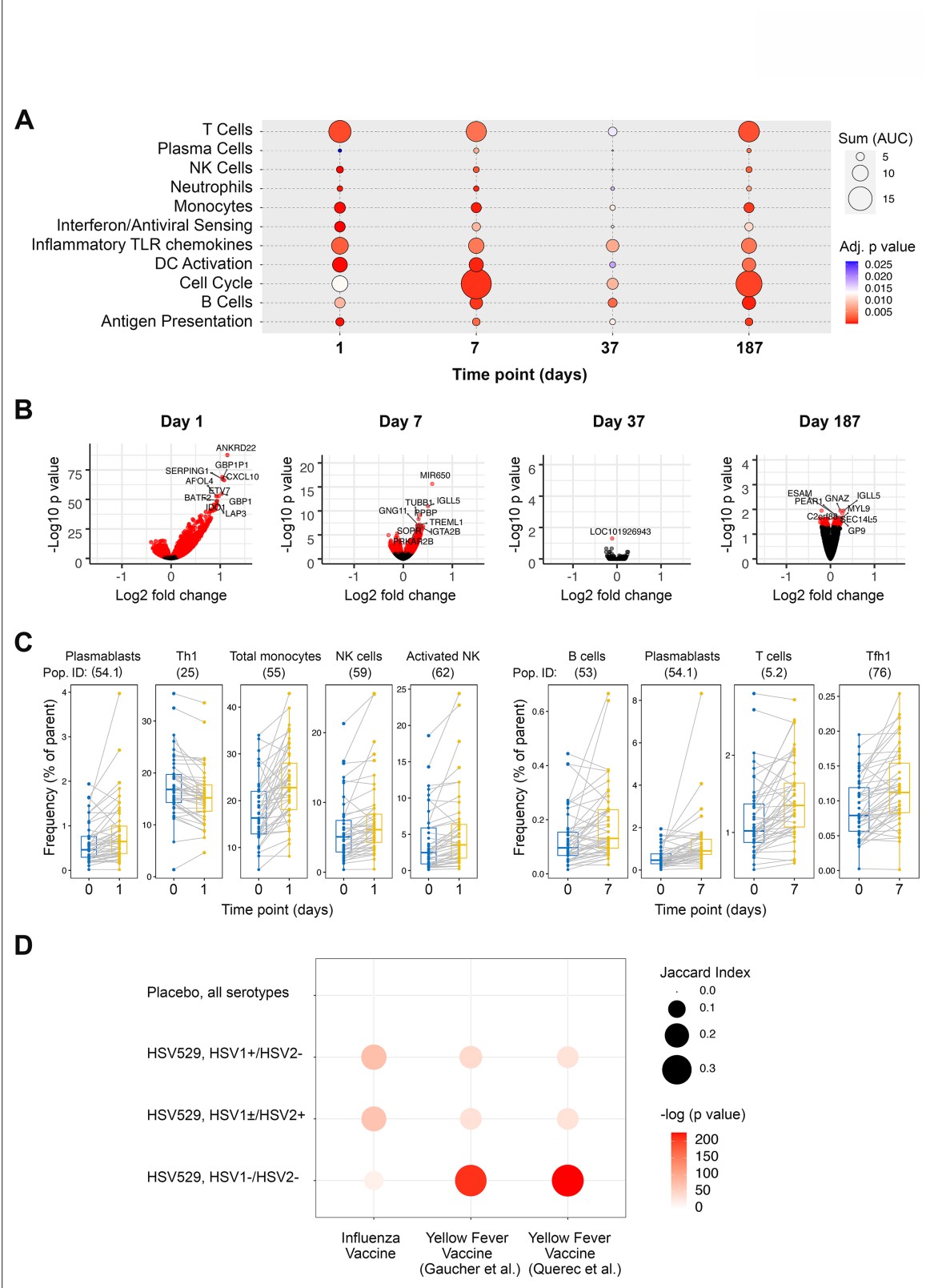

**Figure 2.** Peripheral blood phenotypic responses are greatest on day 1 after vaccination with HSV529 and differ based on prior exposure to herpes simplex virus (HSV). (**A**) Transcriptomic phenotypes of PBMCs were characterized by RNA-seq and responses for all HSV529 recipients were defined by comparing observations at day 1 or 7 to day 0 (day of the first vaccination dose), and for subsequent doses by comparing between the day of the second or third vaccination dose and timepoints 7 days later (days 37 and 187). Functions represented by the genes differentially expressed were

*Figure 2 continued on next page*

*Figure 2 continued*

assessed by enrichment of blood transcription modules (BTMs), and all significantly enriched modules are shown (FDR adjusted p < 0.05) with plotted size and color indicating the normalized effect size and significance, respectively, for enrichments. (**B**) Volcano plots show the changes in individual genes for all HSV529 recipients, with significantly differentially expressed genes colored red (FDR adjusted p < 0.05), and the most significantly differentially expressed genes labeled. Note that the scale of the y axes differ between the days that were analyzed. (**C**) Flow cytometry was used to quantify the frequencies of 70 PBMC populations for all HSV529 recipients. Examples of populations that showed significant changes at either day 1 or 7 after the first vaccine dose compared to day 0 are shown (FDR adjusted p < 0.05). (**D**) Changes in gene expression between days 0 and 1 were next determined separately for subjects that received HSV529 and differed in their status in terms of prior exposure to HSV, or that received placebo and in which all groups of prior exposure were combined (*y*-axis). Jaccard index values and statistical significance, indicated by plotted size and color, respectively, are shown for overlap of these expression changes with the differentially expressed genes reported in previous studies of vaccine responses at day 1 to influenza (*Tsang et al., 2014*) and in two studies of yellow fever (*x*-axis) (*Gaucher et al., 2008*; *Querec et al., 2009*).

The online version of this article includes the following figure supplement(s) for figure 2:

**Figure supplement 1.** Characterization of pathways enriched for overlap between responses to HSV529 and to vaccines for other pathogens.

T follicular helper cells, T cells, and B cells were all significantly expanded by day 7 consistent with an activated adaptive response. Plasmablasts, defined as CD19$^+$ CD20$^-$ CD27$^+$ CD38$^{high}$ (*Perez-Andres et al., 2010*), were significantly expanded at both days 1 and 7 (Wilcoxon's signed-rank paired test with Benjamini–Hochberg correction for multiple comparisons, p < 0.05) (*Figure 2C*). Although this analysis combined all subjects, the remarkably early rise in plasmablasts after vaccination may be related to the particularly rapid neutralizing antibody response to vaccination in HSV seronegative women described above.

The number of genes showing significant changes (FDR corrected p < 0.05) in response to vaccination was higher at 1 day after the first dose of vaccine than at later timepoints and after subsequent vaccine doses (*Figure 2B*). This was also true for the changes in cell populations and might be expected for a replication-defective vaccine which presents antigens to the immune system immediately after vaccination, but is not amplified over time like a replication-competent attenuated virus or vector-based vaccine. Therefore, we further focused analyses on day 1 after the first dose of vaccine, to investigate how subjects differing in sex or prior exposure vary in their responses to HSV529 vaccination.

## Innate responses to HSV529 that differ based on prior exposure resemble differences between innate responses to yellow fever and influenza vaccinations

We analyzed the differentially expressed genes (DEGs) observed after HSV529 vaccination to investigate the transcriptomic responses which were associated with prior exposure to HSV. We compared day 1 DEGs in each of the three HSV serogroups that received HSV529 vaccine, with the early transcriptomic responses that have been previously reported in recipients of either yellow fever or influenza vaccines (*Tsang et al., 2014*; *Gaucher et al., 2008*; *Querec et al., 2009*). In the previous studies, yellow fever is a pathogen for which vaccine recipients had no prior exposure, whereas all subjects receiving the influenza vaccine were expected to have some degree of immunity due to prior influenza infection and vaccine exposure. Jaccard index values, reflecting the size of the intersection relative to the size of the union for two sample sets, were determined to quantify the overlap in day 1 transcriptomic responses to these different vaccines (*Figure 2D*). DEGs in HSV seronegative HSV529 recipients were strikingly similar to those seen in yellow fever vaccine recipients (p < 1 × 10$^{-90}$, Jaccard index >0.35). In contrast, DEGs in both HSV seropositive groups were more similar to those seen in influenza vaccine recipients (p < 1 × 10$^{-30}$, Jaccard index >0.13). Thus, transcriptomic responses observed in HSV naive and seropositive subjects mirror the responses to yellow fever and influenza vaccine recipients, respectively. This indicates that the effect of prior HSV exposure on response to HSV529 is consistent with differences observed between the early responses to vaccines for other pathogens that represent either naive responses or those mounted by individuals with existing adaptive immunity.

Network analysis of pathways that were enriched in overlap between HSV seronegative HSV529 recipients and yellow fever vaccine recipients, showed central roles for gene sets involved in type I IFN (IFN-α/β) and antiviral innate responses (*Figure 2—figure supplement 1A*). In contrast, network analysis of pathways that were enriched in overlap between HSV2 seropositive HSV529 recipients and influenza vaccine recipients, showed a more heterogenous pattern of relatedness that included type II IFN (IFN-γ) and IL-12 responses (*Figure 2—figure supplement 1B*). The 20 most significantly

**Table 1.** Twenty most significantly upregulated genes at day 1 for HSV529 recipients divided into three groups based on prior exposure to herpes simplex virus (HSV).

Changes in expression compared to day 0 were analyzed using Bioconductor package DESeq2, with log 2 fold changes reported and p values corrected for multiple comparisons. Genes marked in red are present in both HSV1$^+$/HSV2$^-$ and HSV1$^\pm$/HSV2$^+$, but not HSV1$^-$/HSV2$^-$ subjects.

| HSV1$^-$/HSV2$^-$ | | | HSV1$^+$/HSV2$^-$ | | | HSV1$^\pm$/HSV2$^+$ | | |
|---|---|---|---|---|---|---|---|---|
| Gene | Fold change | p adjusted | Gene | Fold change | p adjusted | Gene | Fold change | p adjusted |
| MX1 | 1.95 | $7.02 \times 10^{-31}$ | GBP2 | 1.14 | $1.36 \times 10^{-20}$ | ANKRD22 | 5.44 | $4.37 \times 10^{-44}$ |
| EIF2AK2 | 1.28 | $3.99 \times 10^{-28}$ | GBP1 | 2.06 | $2.60 \times 10^{-19}$ | GBP5 | 1.94 | $6.39 \times 10^{-41}$ |
| IFI44 | 2.01 | $3.99 \times 10^{-28}$ | PSTPIP2 | 0.98 | $4.27 \times 10^{-18}$ | WARS | 2.19 | $2.03 \times 10^{-40}$ |
| ISG15 | 2.10 | $3.99 \times 10^{-28}$ | ANKRD22 | 3.54 | $7.25 \times 10^{-18}$ | ETV7 | 4.12 | $5.08 \times 10^{-40}$ |
| IFI6 | 2.26 | $4.94 \times 10^{-28}$ | FCGR1B | 2.32 | $4.39 \times 10^{-17}$ | GBP1 | 2.86 | $1.64 \times 10^{-39}$ |
| HERC5 | 1.66 | $3.52 \times 10^{-27}$ | WARS | 1.44 | $9.49 \times 10^{-17}$ | GBP4 | 1.86 | $1.93 \times 10^{-38}$ |
| OAS3 | 1.95 | $3.52 \times 10^{-27}$ | GBP4 | 1.24 | $2.28 \times 10^{-16}$ | IDO1 | 4.02 | $4.50 \times 10^{-38}$ |
| CMPK2 | 2.12 | $5.75 \times 10^{-27}$ | LAP3 | 1.54 | $2.28 \times 10^{-16}$ | CD274 | 1.72 | $2.16 \times 10^{-37}$ |
| HERC6 | 1.02 | $5.75 \times 10^{-27}$ | GBP5 | 1.24 | $3.51 \times 10^{-16}$ | LAP3 | 2.26 | $7.50 \times 10^{-37}$ |
| IFI44L | 2.68 | $1.52 \times 10^{-26}$ | GBP1P1 | 3.09 | $5.37 \times 10^{-16}$ | GBP2 | 1.46 | $1.71 \times 10^{-36}$ |
| LAMP3 | 1.94 | $2.86 \times 10^{-26}$ | FCGR1A | 2.05 | $7.64 \times 10^{-16}$ | GBP1P1 | 4.40 | $6.42 \times 10^{-34}$ |
| IFIT1 | 2.92 | $1.39 \times 10^{-25}$ | STAT1 | 1.37 | $1.27 \times 10^{-15}$ | VAMP5 | 1.63 | $6.16 \times 10^{-33}$ |
| IFIT2 | 2.41 | $5.10 \times 10^{-25}$ | APOL4 | 3.12 | $2.45 \times 10^{-15}$ | CXCL10 | 4.50 | $1.98 \times 10^{-32}$ |
| SPATS2L | 1.32 | $6.94 \times 10^{-25}$ | PSME2 | 0.84 | $3.67 \times 10^{-15}$ | FCGR1B | 3.10 | $1.98 \times 10^{-32}$ |
| USP18 | 1.89 | $1.30 \times 10^{-24}$ | FCGR1C | 2.55 | $5.59 \times 10^{-14}$ | SERPING1 | 3.79 | $2.05 \times 10^{-32}$ |
| PARP12 | 0.72 | $1.86 \times 10^{-24}$ | IRF1 | 0.89 | $1.52 \times 10^{-13}$ | PSME2 | 1.20 | $2.41 \times 10^{-32}$ |
| RSAD2 | 2.29 | $1.86 \times 10^{-24}$ | APOL2 | 0.64 | $2.19 \times 10^{-13}$ | BATF2 | 3.81 | $2.58 \times 10^{-32}$ |
| TRIM22 | 0.78 | $2.45 \times 10^{-24}$ | PARP9 | 1.14 | $3.14 \times 10^{-13}$ | IRF1 | 1.34 | $3.16 \times 10^{-32}$ |
| HELZ2 | 1.16 | $2.23 \times 10^{-23}$ | APOL3 | 0.55 | $3.89 \times 10^{-13}$ | APOL3 | 0.83 | $7.00 \times 10^{-32}$ |
| MX2 | 1.25 | $5.10 \times 10^{-23}$ | SAMD4A | 1.24 | $3.89 \times 10^{-13}$ | SAMD4A | 1.86 | $1.74 \times 10^{-31}$ |

upregulated genes for HSV529 recipients in each of the 3 HSV serogroups emphasize the similarities in the two HSV2 seropositive groups and how different they are from the seronegative group (*Table 1*). GBP1, WARS, and IRF1, which are upregulated predominantly by IFN-γ (*Megger et al., 2017*; *Sen et al., 2018*) were among the genes most significantly upregulated in both of the HSV seropositive groups studied, while MX1 and ISG15, which are upregulated primarily by IFN-α (*Megger et al., 2017*; *Taylor et al., 1996*), were among the most significantly upregulated genes in HSV seronegative subjects.

There were also differences based on prior exposure to HSV in the cell population responses quantified by flow cytometry. HSV1$^-$/HSV2$^-$ vaccine recipients had significant changes in numbers of activated B cells, Th1, Tc1, and Tc17 cells on day 1, while these cell types were not significantly altered in HSV1$^+$/HSV2$^-$ or HSV1$^\pm$/HSV2$^+$ vaccine recipients (*Supplementary file 1E*). This is consistent with the more robust responses in HSV naive subjects, but it was unexpected that adaptive lymphocyte populations would be activated as early as day 1. However, some cellular responses such as increases in activated NK cells were significantly increased on day 1 in vaccine recipients regardless of their HSV serostatus prior to vaccination. Although naive and memory B cells were monitored these were not among the populations observed to be significantly different, even at day 7, when either HSV1$^-$/HSV2$^-$ or HSV seropositive vaccine recipients were analyzed separately (*Supplementary file 1F*).

## Gene expression responses associated with sex differ significantly between subjects previously naive or exposed to HSV

We next analyzed the transcriptomic changes after HSV529 vaccination to investigate responses associated with sex. Sex had showed a notable effect of faster neutralizing antibody responses in women compared to men within the HSV seronegative group (*Figure 1B*). We hypothesized that differences in early innate responses had contributed to this effect, so we compared transcriptomic responses at day 1 between men and women who were HSV1⁻/HSV2⁻ prior to vaccination, using LIMMA *Smyth, 2004* followed by ranking of genes by the fold change in expression between sexes to analyze enrichment of BTMs (*Li et al., 2014*). Multiple pathways showed significant enrichment with modules including IFN responses, monocytes, and dendritic cell activation which were all upregulated in women, and T cell responses which were increased more in men (FDR adjusted p < 0.05) (*Figure 3A* and *Figure 3— figure supplement 1*). The IFN and antiviral sensing module showed strong enrichment and for each of the six enriched gene sets from this module, all leading-edge genes that drove enrichment are indicated, with the change in expression at day 1 shown for all 15 HSV naive subjects (*Figure 3B*). Interestingly this revealed some heterogeneity, with most but not all women demonstrating stronger upregulation of IFN responses compared to men within HSV naive vaccine recipients. The prominent increase in the day 1 response of HSV naive women is shown for one example of these leading-edge genes, TLR7, which is of particular interest for these sex-associated effects as this gene is encoded on the X chromosome. This response was also vaccine specific as there were no responses in placebo recipients (*Figure 3—figure supplement 2*).

To further assess whether the day 1 responses associated with sex depended on prior exposure to HSV, responses were next compared between men and women, and between each of the three HSV serogroups. Changes in gene expression associated with sex in each serogroup were used to determine differences between the HSV naive group and each HSV exposed group for every gene. This enabled two parallel enrichment analyses; all BTMs that were enriched and showed greater sex-associated differences in the HSV naive compared to exposed groups are shown (FDR adjusted p < 0.05) (*Figure 4A* and *Figure 4—figure supplement 1*). Similar pathways, including IFN responses and dendritic cell activation, were enriched when the effect of sex in HSV naive subjects was compared to its effect in either HSV seropositive group (*Figure 4A*); these pathways were also similar to those enriched when comparing women to men in HSV1⁻/HSV2⁻ vaccine recipients (*Figure 3A*). These results confirm that the effect of sex on transcriptional response to HSV529 vaccination differs significantly between naive and HSV-exposed individuals.

## The transcriptional response in HSV naive women is dominated by IFN responses and correlates negatively with vaccine-induced neutralizing antibody titer

Multiple transcriptional pathways showed enrichment when responses at day 1 were analyzed for a greater effect of sex in HSV naive compared to HSV exposed subjects. These pathways included dendritic cell activation, antigen presentation, monocytes, and neutrophils which were all related to the most significantly enriched module involving IFN antiviral sensing (*Figure 4A*). Therefore, we next focused within the IFN antiviral sensing module, on the seven component gene sets that were each significantly enriched. Examining the leading-edge genes that drove enrichment of these gene sets in individual vaccine recipients revealed that while many of these genes were induced at day 1 in some men or HSV seropositive women, their trend of increasing expression occurred more frequently and consistently in HSV naive women (*Figure 4B*). When these leading-edge genes that drove IFN enrichment were used for principal component analysis of all timepoints and subjects studied, the plot of principal component 1 highlights how distinct day 1 IFN responses were for HSV seronegative women (*Figure 4C*). Thus, the major differences in responses that were shown by women compared to men in the HSV seronegative group, in IFN signatures, were also the major differences when effects of sex were compared between HSV naive and exposed subjects.

The day 1 transcriptional response associated with HSV naive women may have influenced their faster neutralizing antibody responses, which were observed more frequently in the first 30 days after vaccination in comparison to HSV naive men (*Figure 1B*). To test this, variation in magnitude of day 1 IFN responses was assessed for correlation with the early antibody titers induced for HSV naive women. GSVA quantified subject-level variation in antiviral IFN signature responses, which was the

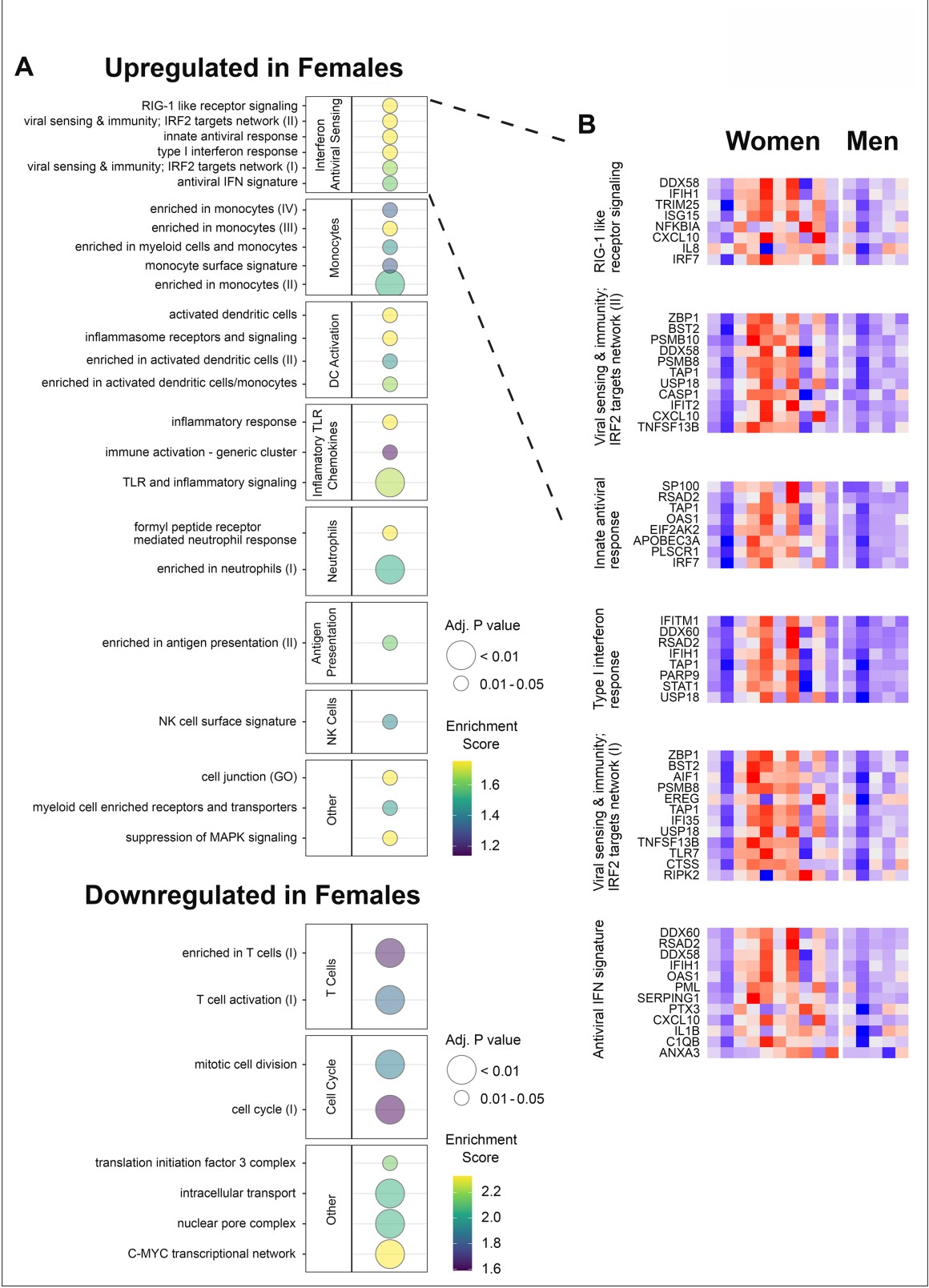

**Figure 3.** Gene expression responses to HSV529 show associations with sex in herpes simplex virus (HSV) seronegative vaccine recipients. (**A**) Changes in expression of all genes between days 0 and 1 were compared between men and women for HSV1⁻/HSV2⁻ subjects, and used to rank genes by fold change for hypergeometric enrichment analysis of blood transcription modules (BTMs). All pathways with significant enrichment in either sex are shown (FDR adjusted p < 0.05), with enrichment score and significance indicated by plotted color and size, respectively. (**B**) For the interferon and antiviral

*Figure 3 continued on next page*

*Figure 3 continued*

sensing module, responses of the 40 different leading-edge genes in 6 represented gene sets are shown in rows for the 15 HSV naive subjects in columns. The change in expression at day 1 compared to day 0 is shown with *z*-scores normalized separately for each of the six gene sets, where red indicates above and blue indicates below mean values.

The online version of this article includes the following figure supplement(s) for figure 3:

**Figure supplement 1.** Enrichment of interferon (IFN) module gene sets in association with sex and prior exposure in day 1 transcriptional responses to HSV529.

**Figure supplement 2.** TLR7 gene expression response to vaccination.

gene set from the IFN and antiviral sensing module that was most significantly enriched in transcriptional responses by HSV naive women (*Figure 4A*; *Hänzelmann et al., 2013*).

Variation in antiviral IFN signature responses at day 1 showed significant negative correlations with HSV2 neutralizing antibody titers, observed either at day 30 or 60, for HSV naive women (Pearson coefficients −0.7, p < 0.04) (*Figure 4D*). Negative correlations for HSV naive women were also seen between HSV2 neutralizing antibody titers and the variation in each of the next five most significant gene sets from the IFN and antiviral sensing module that were enriched in day 1 responses by HSV naive women (*Figure 4—figure supplement 2*). This suggests that the IFN-related transcriptional response that showed the greatest average increase in HSV naive women following vaccination, was associated with lower neutralizing antibody titers within the same group of HSV naive women.

## Discussion

To understand the interaction between sex and prior exposure in the context of an immune response in vivo, we studied the response to a replication-defective HSV2 vaccine in persons with different prior exposure to HSV. We used a systems immunology approach characterizing peripheral blood transcriptomic and cell population phenotypes. While prior studies using systems approaches have been reported for live attenuated virus, subunit, and vectored vaccines, this is the first such study of a replication-defective vaccine in humans. After a longitudinal analysis throughout the three-dose vaccine regimen, we focused on transcriptional responses at day 1 after the first dose, as these exhibited the most robust changes associated with both sex and prior exposure. Changes in these phenotypes were determined and comparisons were made with other outcomes, such as the observation of more robust vaccination responses in women than men in subjects previously naive to HSV, revealed by faster neutralizing antibody responses. Thus, we provide insights for how prior exposure and sex interact to shape the early innate immune response.

The day 1 transcriptional responses differed between HSV529 vaccine recipients based on prior exposure to HSV. Responses to HSV529 by HSV seronegative individuals overlapped strongly with those previously reported for recipients of the yellow fever vaccine, whereas responses to HSV529 by HSV seropositive individuals were significantly different from those who were HSV seronegative and instead overlapped with persons who had been vaccinated with influenza. These findings support the notion that prior exposure to the pathogen against which the vaccine is targeting is a major determinant of the early innate immune response to the vaccine, independent of the vaccine type (live attenuated, subunit, or replication-defective) or the pathogen itself (yellow fever virus, influenza virus, or HSV).Interestingly, IFN responses appeared to be skewed toward IFN-α in the HSV seronegative vaccine recipients but toward IFN-γ in the HSV seropositive group. Type I IFNs, including IFN-α, are produced very early after viral infections, including with HSV, and cellular RNA and DNA sensors for HSV produce type I IFNs in response to virus infection (*Oh et al., 2016*; *Danastas et al., 2020*). IFN-γ signals through a different receptor and is predominantly produced by natural killer cells and T cells during infection (*Platanias, 2005*). IFN-γ is particularly important for inhibiting HSV reactivation and thus may be more prominent when the immune system has previously been exposed to the virus (*Decman et al., 2005*). Both IFNs contribute to the induction of an antiviral state through the upregulation of molecules that can antagonize virus replication, but are increasingly being recognized to exert regulatory effects on the innate response that influence development of adaptive immunity (*Lee and Ashkar, 2018*).

Sex was associated with day 1 transcriptional responses, but the enriched pathways differed between HSV naive and seropositive individuals. A transcriptional profile of increased responses was observed

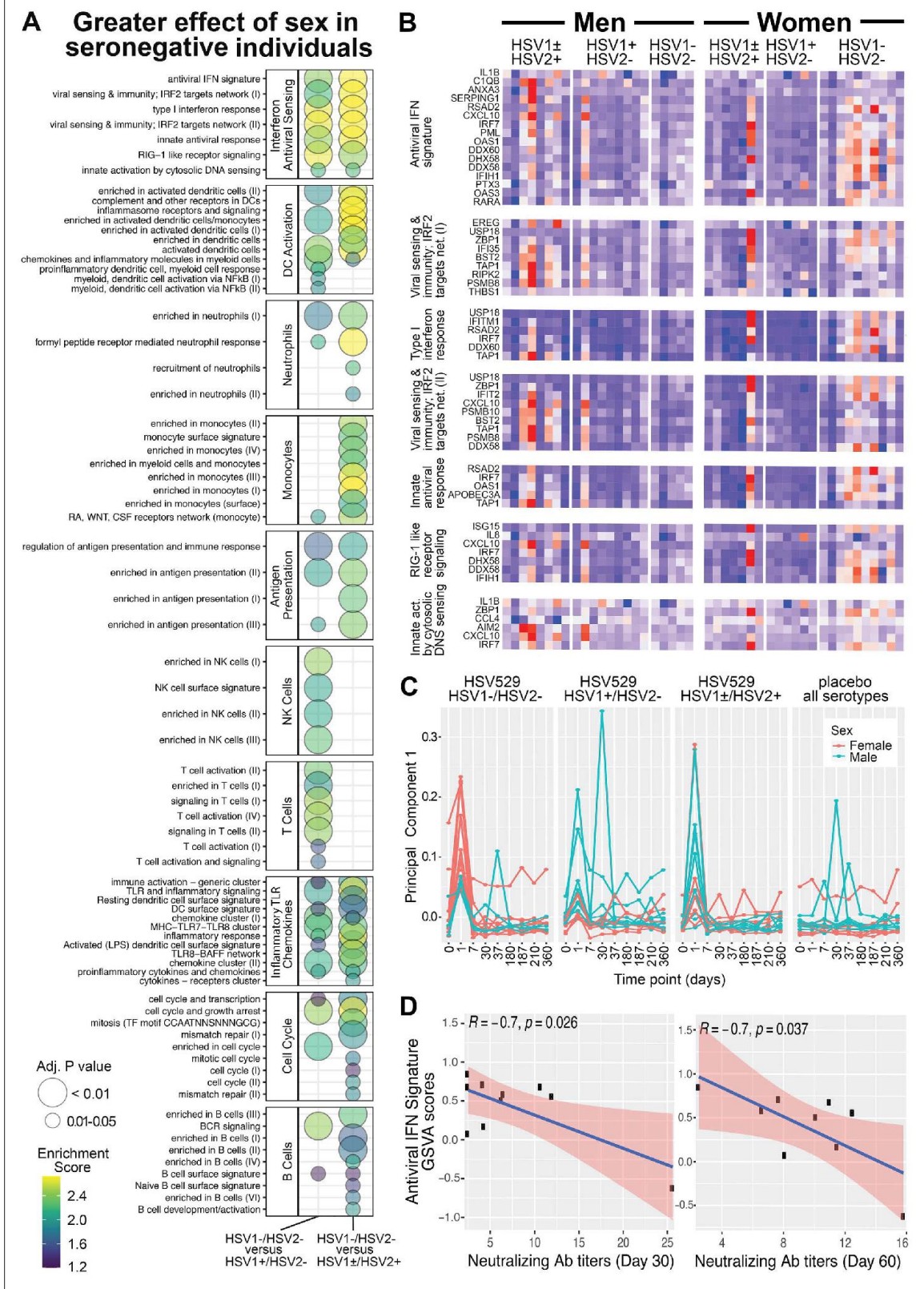

**Figure 4.** Gene expression responses to HSV529 that associate with sex differ significantly between subjects previously naive or exposed to herpes simplex virus (HSV), and the interferon (IFN) response associated with HSV naive women correlates inversely with HSV2 neutralizing antibody titers. (**A**) Changes in expression of all genes between days 0 and 1 were compared between men and women separately for the three HSV serogroups of subjects based on prior exposure to HSV. For each gene those changes with sex were then used to compute a difference between the HSV naive group and

*Figure 4 continued on next page*

*Figure 4 continued*

either HSV exposed group (*x*-axis). These differences were used to rank genes for hypergeometric enrichment analysis of blood transcription modules (BTMs), and all pathways with significant positive enrichment in the HSV naive group are shown (FDR adjusted p < 0.05), with enrichment score and significance indicated by plotted color and size, respectively. (**B**) Comparison of sex differences between HSV naive and HSV1⁺/HSV2⁻ subjects identified seven gene sets significantly enriched in the IFN antiviral sensing module. For these seven gene sets, responses of the 33 different leading-edge genes in rows, for all 45 vaccine recipients in columns are shown with the change in expression at day 1 compared to day 0 with *z*-scores normalized within each gene set, where red indicates above and blue indicates below mean values. (**C**) Principal component analysis was performed using expression of the 33 leading-edge genes from panel B, at all nine timepoints with RNA-seq data. The first principal component is plotted for all 60 subjects colored by sex and separated by HSV serostatus for vaccine recipients, or with all serogroups combined for placebo recipients. (**D**) For HSV naive women that were vaccinated, variation in day 1 IFN responses was quantified by gene set variation analysis (GSVA) of the antiviral IFN signature (*y*-axis), and correlated with HSV2 neutralizing antibody titers observed at day 30 or 60 (*x*-axis). Pearson coefficient and significance values are shown for the linear correlation in blue, with 95% confidence interval shaded red.

The online version of this article includes the following figure supplement(s) for figure 4:

**Figure supplement 1.** Enrichment of interferon (IFN) module gene sets in association with sex and prior exposure in day 1 transcriptional responses to HSV529.

**Figure supplement 2.** Multiple interferon (IFN)-related gene sets enriched in day 1 responses by herpes simplex virus (HSV) naive women correlate inversely with HSV2 neutralizing antibody titers.

for women, specifically for those previously naive to HSV. The transcriptomic modules and leading-edge genes that characterized this response were centered on type I IFN and antiviral pathways. This suggests that in the absence of prior exposure, the more robust transcriptional responses in women to a replication-defective virus may be due to an underlying difference in baseline inflammation or in the speed of induction of inflammation in comparison to men. Such IFN-associated broad immune activation has recently been found to underlie shared set point signatures for increased vaccine responsiveness in healthy individuals of both sexes (although this signature tends to be more elevated in females than males at baseline), as well as disease activity in patients with lupus (**Kotliarov et al., 2020**). Thus, the more robust day 1 transcriptional responses in HSV naive women could have contributed to the more rapid neutralizing antibody responses observed in HSV naive women in this cohort. Interestingly, we found that within HSV naive women the transcriptional IFN response correlated negatively with vaccine-induced antibody titers. Possibly, as HSV529 is a replication-defective vaccine dependent on an initial infection of host cells, early IFN responses inhibited the number of cells infected or the efficiency of viral antigen expression. This is consistent with the established role of IFN antagonizing natural HSV infection (**Leib, 2002**) and indicates how the effects of IFN responses on vaccine effectiveness may be context dependent. Further work will be necessary to assess whether similar observations can be replicated in other naive responses to replication-defective or live attenuated vaccines, including similar consequences for the induction of antibody titers.

Despite the IFN response on day 1 that was associated with reduced antibody titers in HSV naive women, these individuals as a group showed the most robust antibody responses to HSV529, with more rapid antibody responses compared to HSV naive men. Overall, HSV naive individuals showed greater titer increases compared to HSV exposed subjects particularly at later timepoints. The diminished responses by HSV exposed subjects may also have resulted from suppression of HSV529 infection, in this case by preexisting neutralizing antibody or T cell immunity. This is supported by the large number of genes upregulated on day 1 after vaccination in HSV1⁻/HSV2⁻ persons in comparison with the much smaller number of genes upregulated in these same individuals with subsequent doses of vaccine, or in HSV1⁺/HSV2⁻ or HSV1±/HSV2⁺ subjects at any timepoint. Evidence for preexisting antibody inhibiting live virus vaccination has been reported for other pathogens such as measles, for which efficacy improves if vaccination is administered later within the first year of life; clinical data combined with experimental models indicate this is more likely due to interference from maternal antibodies rather than immaturity of the neonatal system (**Aaby et al., 2014**; **van Binnendijk et al., 1997**), although the underlying mechanisms remain largely unclear. Another example is the live attenuated intranasal influenza vaccine which is approved in the United States for persons aged 2–49; prior exposure to influenza may lead to neutralization and reduced replication of the live influenza virus vaccine resulting in reduced efficacy in adults compared to children (**Hoft et al., 2017**; **Monto et al., 2009**; **Grohskopf et al., 2021**). In HSV exposed subjects a higher dose of HSV529 may be required to

achieve a better response; indeed the live attenuated zoster vaccine is given at 14 times the dose of the varicella vaccine, in part, to overcome the effect of preexisting immunity (*Cohen, 2008*).

More robust responses by women compared to men have been described for several vaccines (*Klein et al., 2015*), and our findings demonstrate some overlap with prior studies of sex differences in the context of HSV. In two double-blind randomized studies of an HSV subunit vaccine, the vaccine was protective against genital herpes in women who were HSV1⁻/HSV2⁻ prior to vaccination, but the vaccine was not effective in men (*Stanberry et al., 2002*). A subsequent large double-blind placebo-controlled trial found that women reported more symptoms and developed higher levels of antibody than did men after vaccination; however, the differences in antibody titers were not significant, and a later large study focused on HSV1⁻/HSV2⁻ women failed to replicate efficacy against HSV2 infection or disease (*Bernstein et al., 2005*; *Belshe et al., 2012*). Murine studies have also implicated IFN responses in association with sex in the context of HSV. Male mice have a higher mortality rate after HSV infection than females which correlates with the difference in IFN-γ response, and male IFN-γ knockout mice have higher HSV reactivation rates than control mice whereas female IFN-γ knockout mice show no difference from controls (*Han et al., 2001*). We now show that in the context of human responses to HSV529, sex-associated differences in IFN and inflammatory signaling are significantly more pronounced in women not previously exposed to HSV.

Prior studies suggest that HSV2 subunit vaccines are more effective in women than men (*Stanberry et al., 2002*), but the underlying mechanisms of the sex-dependent response may be different for the live, replication-defective HSV2 vaccine used in our study. For example, HSV is known to activate TLR9, and thus a replication-defective vaccine like ours would be expected to activate TLR9 by presenting viral DNA to the immune system. TLR9 responses are known to be sex dependent, with a role in autoimmune disorders disproportionately affecting females, and may have thus contributed to the more robust innate responses in females in our study (*Lund et al., 2003*; *Marshak-Rothstein, 2006*). A replication-defective live viral vaccine may also be more effective in triggering the presentation of processed viral peptides on the surface of the infected host cells to stimulate innate and virus-specific T cell responses compared to protein subunit vaccines. Thus, our HSV2 replication-defective vaccine may be more likely to trigger innate and cellular immune response pathways that are more robust in women, for example, similar to earlier observations of more robust IFN-α production and CD8 T cell activation after stimulation of PBMCs with a viral peptide in HIV-infected women compared to men (*Meier et al., 2009*).

Due to the more rapid vaccine response in seronegative women observed in this study, we analyzed changes in plasmablasts. In a trial of the malaria recombinant protein vaccine RTS,S, increased plasmablasts were observed on day 1, but only after the second and third doses of vaccine (*Kazmin et al., 2017*). We detected expansion of plasmablasts when analyzing all subjects at both days 1 and 7 after the first dose of vaccine, but these changes did not reach statistical significance when stratifying by prior exposure and sex. We also observed that the numbers of plasmablasts peaked around day 7 as others have reported, although the early response we observed at day 1 was unexpected (*Wrammert et al., 2008*; *Magnani et al., 2017*).

Several individual genes were highlighted by our transcriptomic screen. TLR7 is an X-linked gene for which higher levels of expression have been reported in B cells of naive female versus male mice after vaccination with inactivated influenza vaccine, and these higher levels of TLR7 correlated with high levels of antibodies in the female mice (*Fink et al., 2018*). We also detected higher levels of TLR7 gene expression in HSV1⁻/HSV2⁻ women compared to men on day 1 after vaccination. TLR7 is important for immunoglobulin class switching and could represent a molecular mechanism for sex-based differences, which have been attributed to X-linked immune response genes as well as to hormonal differences (*Harper and Flanagan, 2018*; *Klein and Flanagan, 2016*). A further observation arising from the broad phenotypic characterization is the concordance between significant host gene expression responses and pathogen immune evasion. The most significantly upregulated individual genes were dominated at day 1 by IFN-stimulated (*CXCL10*, *GBP1*, and *IFIT2*) or Fcγ receptor genes (*FCGR1A*, *FCGR1B*, and *FCGR1C*). At day 7, the most significantly upregulated individual genes included integrins (*ITGB3*, *ITGB5*, and *ITGA2B*). HSV encodes immune evasion proteins that antagonize many of these specific gene functions: blocking IFN, antibody-mediated cytolysis, and activity of integrin β3 (*Leib, 2002*; *Lubinski et al., 1998*; *Gianni et al., 2012*).

Overall, our study confirms and adds to the substantial effects observed for prior exposure and sex in innate immune responses. We document at a systems level how these two variables interact in the context of response to a replication-defective vaccine, and determine resulting transcriptional response profiles that may form the basis for differences in neutralizing antibody responses in vivo. To improve vaccination effectiveness, it will thus be important to better understand how prior exposure and sex affect the innate responses that are increasingly being shown to shape subsequent adaptive immunity.

## Acknowledgements

We thank Mirko Trilling for comments and advice about IFN-regulated genes, and Ronald Germain, Neal Young, Giorgio Trinchieri, and Pam Schwartzberg for input on study design. This research was supported by the Intramural Research Program of the NIH, the National Institute of Allergy and Infectious Diseases, and other institutes supporting the Trans-NIH Center for Human Immunology, Autoimmunity, and Inflammation. The vaccine trial was supported through a clinical trial agreement between the National Institute of Allergy and Infectious Diseases and Sanofi Pasteur.

## Additional information

### Funding

| Funder | Grant reference number | Author |
|---|---|---|
| Division of Intramural Research, National Institute of Allergy and Infectious Diseases | | Foo Cheung<br>Richard Apps<br>Lesia Dropulic<br>Yuri Kotliarov<br>Jinguo Chen<br>Tristan Jordan<br>Marc Langweiler<br>Julian Candia<br>Angelique Biancotto<br>Kyu Lee Han<br>Nicholas Rachmaninoff<br>Harlan Pietz<br>Kening Wang<br>John S Tsang<br>Jeffrey I Cohen |

The funders had no role in study design, data collection, and interpretation, or the decision to submit the work for publication.

### Author contributions

Foo Cheung, Formal analysis, Investigation, Writing – review and editing, Data curation, Methodology, Software, Validation, Visualization; Richard Apps, Investigation, Writing – review and editing, Project administration, Resources, Writing – original draft; Lesia Dropulic, Conceptualization, Investigation, Resources, Writing – review and editing; Yuri Kotliarov, Formal analysis, Methodology, Resources; Jinguo Chen, Investigation, Methodology, Validation; Tristan Jordan, Marc Langweiler, Angelique Biancotto, Kyu Lee Han, Investigation; Julian Candia, Formal analysis, Methodology; Nicholas Rachmaninoff, Harlan Pietz, Resources; Kening Wang, Investigation, Resources; John S Tsang, Conceptualization, Supervision, Project administration, Writing – review and editing, Funding acquisition, Methodology, Resources, Writing – original draft; Jeffrey I Cohen, Conceptualization, Funding acquisition, Writing – original draft, Project administration, Resources, Supervision, Writing – review and editing

### Author ORCIDs

Richard Apps ⓘ http://orcid.org/0000-0001-5140-0141
Julian Candia ⓘ http://orcid.org/0000-0001-5793-8989
Harlan Pietz ⓘ http://orcid.org/0000-0003-4792-5708
John S Tsang ⓘ http://orcid.org/0000-0003-3186-3047
Jeffrey I Cohen ⓘ http://orcid.org/0000-0003-0238-7176

### Ethics

Clinical trial registration clinicaltrials.gov ID: NCT01915212.

All HSV529 vaccine recipients signed informed consent for a protocol (clinicaltrials.gov ID: NCT01915212) approved by the National Institute of Allergy and Infectious Diseases Institutional Review Board.

### Decision letter and Author response

Decision letter https://doi.org/10.7554/eLife.80652.sa1
Author response https://doi.org/10.7554/eLife.80652.sa2

---

## Additional files

### Supplementary files

• Supplementary file 1. Flow cytometry markers and lists of significantly upregulated genes and changes in cell populations. (A) Four parallel 10-color staining panels used for high parameter flow cytometry analysis. (B) Seventy subsets of PBMCs detected by high parameter flow cytometry analysis. (C) The 20 most significantly upregulated genes at day 1 compared to day 0 when analyzing all HSV529 recipients. Genes marked in red are IFN-stimulated genes, and those in blue are Fc receptors. (D) The 20 most significantly changing cell populations at day 1 compared to day 0 when analyzing all HSV529 recipients. Red indicates NK cells, blue indicates Th1 cells, green indicates cytolytic cells and yellow indicates plasmablasts. (E) Significantly changing cell populations in HSV529 recipients based on HSV serogroup, when analyzing differences from days 0 to 1. (F) Significantly changing cell populations in HSV529 recipients based on HSV serogroup, when analyzing differences from days 0 to 7.

• Transparent reporting form

### Data availability

Microarray data, cell population frequencies, and neutralizing antibody titers are available at GEO (accession number GSE185341). Transcriptome data for 9 of the 60 vaccine recipients obtained at days 180 and 187 after immunization is excluded from the GEO database since blood was drawn on these participants at those timepoints after genomic data sharing language was revised and this new language was not added to the participant's consent forms.

The following dataset was generated:

| Author(s) | Year | Dataset title | Dataset URL | Database and Identifier |
|---|---|---|---|---|
| Cheung F, Apps R, Dropulic L, Kotliarov Y, Chen J, Jordan T, Langweile M, Candia J, Biancotto A, Han KL, Rachmaninoff N, Pietz H, Wang K, Tsang SJ, Cohen JI | 2023 | Sex and prior exposure jointly shape innate immune responses to a replication-defective herpesvirus vaccine | https://www.ncbi.nlm.nih.gov/geo/query/acc.cgi?acc=GSE185341 | NCBI Gene Expression Omnibus, GSE185341 |

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
