## [Editor Report]

This important study uses a systems immunology approach to disentangle the effect of sex and prior exposure on an individual's response to viral vaccines. This work brings an impressive data assortment and present compelling evidence to support the conclusions. The paper would be of interest to researchers working in human immunology and vaccine development.

---

## [Decision Letter]

**Decision letter after peer review:**

Thank you for submitting your article "Sex and prior exposure jointly shape innate immune responses to a replication-defective herpesvirus vaccine" for consideration by *eLife*. Your article has been reviewed by 3 peer reviewers, and the evaluation has been overseen by a Reviewing Editor and Arturo Casadevall as the Senior Editor. The following individuals involved in the review of your submission have agreed to reveal their identity: Assya Trofimov (Reviewer #1); Sita Awasthi (Reviewer #3).

Essential revisions:

1. If data is available, the authors should check the enrichment of TLR7 (or lack thereof) in the placebo group to rule out the potential vaccine-driven effects.

2. Extend the discussion on the BTM analysis in the Methods section. More details are necessary.

3. Discuss the possibility of sex-specific variations in immunological outcomes after vaccination with a replication-defective virus compared to an adjuvanted subunit vaccine.

4. Modify line 162 (or Figure S1) so the message is consistent.

5. Please follow reviewers' suggestions to improve the readability of figures throughout the manuscript.

6. Please correct the typos listed by the reviewers.

*Reviewer #1 (Recommendations for the authors):*

Sex and prior exposure to the antigen are known factors that influence the immune response to vaccines. In this study, the authors study how the combination of these factors changes the immune response to a multi-dose replication-defective herpes simplex virus (HSV) vaccine.

Strengths

The main strength of this study is the variability in the data that was acquired and the granularity of the annotation. The authors meticulously established comparisons between the vaccine recipient subgroups and came up with interesting results and conclusions.

Weakness

Some analyses were challenging to understand and the methods section would have benefited from additional details.

I would have wanted to see a more robust conclusion about the loss of sex-specific differences in antigen-experienced individuals compared to naive individuals. This conclusion is alluded to but not explicitly made.

The authors achieved their aims in quantifying the difference in the immune response between naive individuals and those with previous exposure to HSV as well as differences between males and females. Their results fully support the conclusions stated in the manuscript.

This manuscript provides a comprehensive assessment of the differences between male and female responders as well as naive and virus-experienced responders. These results show many interesting leads for a better understanding of these differences. These results are especially important in the context of the development of vaccines against viruses that establish lifelong infections, such as HSV, CMV and VZV. Since the authors have committed to releasing the data, this will likely offer other teams some rich data for subsequent analysis.

I. Comments and questions:

a) Analysis for Figures B and C: Since all of the Δ_Day0 Neutralizing Ab Titers are at zero on Day 0, I'm not sure the statistical test is accurate. The test compares means and since all values have a mean of zero on the first day (post normalization) and necessarily do not have a mean of zero at the next time points (as per the normalization procedure), this makes me conclude that the test is likely biased. Perhaps changing the strategy would be best. The authors seem to want to verify if the difference between men and women is significant. Perhaps simply comparing the values at the second timepoint would be a better use of a statistical test and reflect reality. So my suggestion is:

ttest(a, b) where a = women's normalized titer @ 30 Days and b = men's normalized titer @ 30 Days.

b) Line 162: This statement is not entirely true. Since Supplementary Figure 1 only shows recorded responses for HSV- individuals (as per legend description), only based on that figure it is impossible to state that prior exposure markedly affects the response to the vaccine. These results are not even in that Figure! If they are, the legend should be adjusted, if not, they should be added to match the conclusion!

c) Figure 2, 3, 4: For each of these Figures, the manuscript would benefit from clear indications of which subgroups are being compared.

d) Methods should contain a clear description of the BTM analysis: how was it done? Which software was used? What are the gene sets used?

e) I am curious about Dose 2. What happened to it? Why is there such a low cell enrichment (Figure 2A) and such a low number of differentially expressed genes (Figure 2B)? Was the immune response sustained from Day 7 to Dose 3 and this is why there are few differentially expressed genes? Was there a problem in the quantification?

f) Why did the authors combine all responders for Figure 2C when they suspected differences in sex and between naive/antigen-experienced groups?

g) I really like the analysis done in Figure 2D. I think the link to other vaccine studies is interesting. Since DEGs are often very dependent on the dataset, perhaps repeating the comparison but with enriched pathways or even the BTMs would be more convincing. I feel like the signal is slightly blurred with the DEGs and Jaccard index since DEGs are often dataset-specific and the Jaccard index is so highly dependent on the overall set size. I suspect the authors will find a greater signal using pathways instead.

h) Line 246: IL-12 has some influence on Th1 adaptive immune response. Perhaps splitting the results from Figure 2C into seronegative and seropositive would have been better and would show the difference in IL-12-based Th1 cell enrichment.

i) Line 300: The authors are correct to note that TLR7 is located on chromosome X. Would it be possible to check within the placebo group comparing males and females if TLR7 is still significantly enriched? This would rule out any vaccine-based effect compared to the sex-based only effect.

j) Figure 4 and Lines 334 and 336. Authors state similar pathway enrichments but then conclude that there is a difference in the effect of sex between seropositive and seronegative groups. In Figure 4B, it seems like the biggest difference is seen between seronegative individuals. And yet there are the same genes enriched in.

k) Figure 4A. Perhaps some additional explanation could reconcile this contradiction in this paragraph!

Line 370: So if I understand correctly, seropositive females lose the high IFN response because of a higher neutralizing titer (as seen in seronegative/positive males?)

l) Data was not available and is currently embargoed until October 2024!

*Reviewer #2 (Recommendations for the authors):*

The authors study responses to a non-replicating human HSV vaccine. They find that the vaccine responses differ in males and females, predominantly in those individuals who are HSV seronegative.

Strengths.

The authors have a relatively large cohort of patients enabling longitudinal studies in seronegative and seropositive male and female individuals

The authors performed a comprehensive study of the immune responses both on the level of neutralizing antibodies and gene activation for long time periods

Their work identified clear differences between males and females in seronegative individuals: females produce neutralizing ab's faster, but males have delayed but higher levels.

The work is able to relate some of the differences to the differences in the Type I IFN response.

Weaknesses.

Many of the observations reported in the paper have already been described in the literature, such as sex-specific immune responses to viral vaccines.

The possible causal connection between the kinetics of gene activation and the differences in ab responses is discussed but requires further work.

The paper leaves uncertain if this vaccine is going to be epidemiologically/clinically useful (as these are Phase 1 clinical trials).

I am uncertain what is the significance of these results for understanding the fundamental biology of the immune responses to vaccination.

I think this is a technically very strong paper that reports interesting results

However, I am not sure it reports a breakthrough in our understanding of either the vaccine responses or the sex-linked differences.

I think this is important work that might be more appropriate in a human immunology/vaccinology journal.

*Reviewer #3 (Recommendations for the authors):*

This study describes early antiviral innate responses of a replication-defective herpesvirus vaccine and the subsequent impact on adaptive responses based on the sex of the vaccinee and prior exposure to the virus. As early as one day after the vaccination the differences depended on sex and prior exposure to the virus were noted using blood transcriptomics and cell population profiling. Type I interferon (IFN) signature was found to be the most pronounced in HSV naive women and associated with lower neutralizing antibody titers, suggesting that a strong early antiviral response reduced the uptake of this replication-defective virus vaccine.

Strengths:

The methodology used by the authors has a wider implication beyond the replication-defective herpes virus vaccine.

This is the first detailed study using the systems immunology approach to assess early immune responses and transcriptomic analysis for a replication-defective herpesvirus vaccine.

The authors have provided a robust data set to justify the conclusions.

Figures are clear and statistical analysis is vigorous.

Weaknesses:

Overall, this manuscript has minimal weaknesses.

The manuscript shows that neutralizing activity is negatively correlated with IFN type 1 signatures in females, it would be of great interest to analyze whether overall gD2 binding or HSV-2 binding antibody levels are also lower in females compared to males or only the antibody with neutralizing activity.

If total binding antibody levels are not different, analyzing the Fc receptor-mediated activity such as ADCC would be of great interest.

A few recommendations are made to improve the manuscript

1. Since this study most likely used the same sets of samples that were used in a prior study (Dropulic et al. JID 2019 published by the same group), it may be possible to discuss whether the difference in binding antibody was noted. Alternatively, the data may be further analyzed by sex for discussion purposes.

2. Day 1 following 1st immunization seems to show the most interesting pattern in HSV naïve population. Conceptually, with a replication-defective viral vaccine, the subsequent immunizations in the HSV naive group are more likely to behave similar to that of the virus-exposed group and consequently, immune responses may be indicative of that. It is not clear whether later time point samples were assessed and analyzed based on the naive and prior exposure group.

3. Volcano plot for figure 2B clearly shows the changes in individual genes in samples collected over time. The Y-axis scale variations among panels minimize the appearance of huge differences. If it is difficult to maintain the same scale for all the panels, a comment can be inserted in the figure legend to note the differences in the scale.

4. Previously published phase III trials with subunit vaccines showed higher efficacy in the HSV naïve women. It is to be noted that higher efficacy in females was not assessed for correlation with neutralizing activity in the prior study. One of the most interesting findings presented here is that a replication-defective vaccine had a smaller boost in neutralizing activity in females following 2nd immunization compared to men. It may be interesting to discuss the possibility of variations in immunological outcome based on the sex with a replication-defective viral vaccine compared to an adjuvanted subunit vaccine.

---

## [Author Response]

Essential revisions:1. If data is available, the authors should check the enrichment of TLR7 (or lack thereof) in the placebo group to rule out the potential vaccine-driven effects.

TLR7 expression is profiled, including for recipients of placebo stratified by gender, in Figure 3—figure supplement 2. This shows that the TLR7 response is not observed in placebo recipients of either gender, and this is now highlighted in the text (line 325).

2. Extend the discussion on the BTM analysis in the Methods section. More details are necessary.

TLR7 expression is profiled, including for recipients of placebo stratified by gender, in Figure 3—figure supplement 2. This shows that the TLR7 response is not observed in placebo recipients of either gender, and this is now highlighted in the text (line 325).

3. Discuss the possibility of sex-specific variations in immunological outcomes after vaccination with a replication-defective virus compared to an adjuvanted subunit vaccine.

We now discuss in a dedicated paragraph in the discussion (lines 466-479), potential mechanisms underlying differences in responses between sexes in the context of a replication-defective versus a subunit protein vaccine. Among those are that TLR9 and CD8 T cell pathways may be driven more strongly by the replication-defective vaccine, and that these have previously been reported to show sex-specific variation.

4. Modify line 162 (or Figure S1) so the message is consistent.

Figure 1—figure supplement 1 has now been modified to include HSV+ in addition to HSV- individuals, and we have modified the text, now lines 220-225. These continue to support that adverse reactions and titer responses are associated with both sex and prior exposure.

5. Please follow reviewers' suggestions to improve the readability of figures throughout the manuscript.

Corrections have been made, such as to the axis of Figures 4C and D, Figure 3—figure supplement 2, and to the legends of Figures2,3,4.

6. Please correct the typos listed by the reviewers.

Corrections have been made such as to the parenthesis on line 55 and text on line 95

Reviewer #1 (Recommendations for the authors):Sex and prior exposure to the antigen are known factors that influence the immune response to vaccines. In this study, the authors study how the combination of these factors changes the immune response to a multi-dose replication-defective herpes simplex virus (HSV) vaccine.StrengthsThe main strength of this study is the variability in the data that was acquired and the granularity of the annotation. The authors meticulously established comparisons between the vaccine recipient subgroups and came up with interesting results and conclusions.WeaknessSome analyses were challenging to understand and the methods section would have benefited from additional details.

See response to essential revisions comments 2 and 5 above.

I would have wanted to see a more robust conclusion about the loss of sex-specific differences in antigen-experienced individuals compared to naive individuals. This conclusion is alluded to but not explicitly made.

See response to essential revision comment 3 above

The authors achieved their aims in quantifying the difference in the immune response between naive individuals and those with previous exposure to HSV as well as differences between males and females. Their results fully support the conclusions stated in the manuscript.This manuscript provides a comprehensive assessment of the differences between male and female responders as well as naive and virus-experienced responders. These results show many interesting leads for a better understanding of these differences. These results are especially important in the context of the development of vaccines against viruses that establish lifelong infections, such as HSV, CMV and VZV. Since the authors have committed to releasing the data, this will likely offer other teams some rich data for subsequent analysis.I. Comments and questions:a) Analysis for Figures B and C: Since all of the Δ_Day0 Neutralizing Ab Titers are at zero on Day 0, I'm not sure the statistical test is accurate. The test compares means and since all values have a mean of zero on the first day (post normalization) and necessarily do not have a mean of zero at the next time points (as per the normalization procedure), this makes me conclude that the test is likely biased. Perhaps changing the strategy would be best. The authors seem to want to verify if the difference between men and women is significant. Perhaps simply comparing the values at the second timepoint would be a better use of a statistical test and reflect reality. So my suggestion is:ttest(a, b) where a = women's normalized titer @ 30 Days and b = men's normalized titer @ 30 Days.

We feel that performing a δ using the baseline at day 0, i.e. (D30-D0) and then analyzing this using Mann-Whitney-Wilcoxon for either males or females separately within the seronegative group is an informative approach to assess whether a more robust titer responses after vaccination are displayed by seronegative females. This showed a significant response in the female group (p<0.001) but insignificant change for males, consistent with the observation that increases in titer at day 30 are seen for 7 out 10 females but only 1 out of the 5 males.

We also performed the test suggested by this reviewer. This specific test yields p=0.1, even though our analysis described above did highlight the possibility of responses distinct to seronegative females. We thus think that we are likely not powered to assess significant differences in titer responses between groups. However, as shown in the manuscript, we subsequently investigated further by analyzing immune phenotypes and showed, at the transcriptional level, significantly different responses in seronegative females compared to males or seropositive subjects (Figures3 and 4). These data together support our overall conclusion.

b) Line 162: This statement is not entirely true. Since Supplementary Figure 1 only shows recorded responses for HSV- individuals (as per legend description), only based on that figure it is impossible to state that prior exposure markedly affects the response to the vaccine. These results are not even in that Figure! If they are, the legend should be adjusted, if not, they should be added to match the conclusion!

We have revised the text, now lines 220-225, as well as included responses for HSV+ individuals in the figure, now Figure 1—figure supplement 1.

c) Figure 2, 3, 4: For each of these Figures, the manuscript would benefit from clear indications of which subgroups are being compared.

We have revised the figure legends to be clear which subgroups are being compared. The legend to Figure 2A states in the first sentence that all groups are being compared (no subgroups) and we have now stated this for the legend for Figure 2B and 2C. Figure 2D indicates we used subgroups based on HSV serology. The first sentence of the legend to figure 3 states that the figure is analyzing HSV seronegative vaccine recipients based on sex. The first sentence of the legend to figure 4 states that the figure is analyzing vaccine recipients based on HSV serology and sex (the bottom of panel A shows the subgroups and the top of panels B and C show the subgroups), while the legend for panel D indicates it shows HV seronegative women.

d) Methods should contain a clear description of the BTM analysis: how was it done? Which software was used? What are the gene sets used?

See response to essential revision comment 2 above.

e) I am curious about Dose 2. What happened to it? Why is there such a low cell enrichment (Figure 2A) and such a low number of differentially expressed genes (Figure 2B)? Was the immune response sustained from Day 7 to Dose 3 and this is why there are few differentially expressed genes? Was there a problem in the quantification?

The immune responses appeared much higher to the first dose of vaccine than to either the 2^nd^ (day 30) or 3^rd^ (day 180) doses. Note the large difference in scale of the y axis in figure 2B day7, 37, and 187 compared to day1, and in response to the reviewers comment we now highlight this in the legend for figure 2B. There were not technical problems when quantifying response to dose 2, although we did not collect blood on days 31 and 188 (day 1 after the 2^nd^ and 3^rd^ doses of the vaccine) and may have missed the peak of those responses. There was also not sustained transcriptomic perturbation at the day of the second does, indicated by very few DEG comparing between day 30 and day 0. Differential responses between doses have been documented for other vaccines, and may reflect altered responsiveness particularly when re-vaccinating within a short period of 30 days (dose 2) compared to 150 days (dose 3).

f) Why did the authors combine all responders for Figure 2C when they suspected differences in sex and between naive/antigen-experienced groups?

Despite the potential sex/exposure effects, in Figure 2A-C we initially combined all subjects to first characterize what the vaccine is doing to the immune system in terms of transcriptomic and cell population responses overall. We then go on to explore effects of sex and prior exposure in subsequent figures.

g) I really like the analysis done in Figure 2D. I think the link to other vaccine studies is interesting. Since DEGs are often very dependent on the dataset, perhaps repeating the comparison but with enriched pathways or even the BTMs would be more convincing. I feel like the signal is slightly blurred with the DEGs and Jaccard index since DEGs are often dataset-specific and the Jaccard index is so highly dependent on the overall set size. I suspect the authors will find a greater signal using pathways instead.

We appreciate this suggestion but, given the analysis using DEGs already showed a clear effect, we have not gone on to explore the analysis at the pathway level to avoid inundating the reader with additional data (since this is already an omics level study with a dense presentation of data). We agree it could show a greater signal but feel the important point of overlap with other vaccines based on prior exposure is already shown, and that the particular pathways involved are also well characterized in those prior studies of vaccines in which all recipients are either naïve (yellow fever) or previously exposed (influenza).

h) Line 246: IL-12 has some influence on Th1 adaptive immune response. Perhaps splitting the results from Figure 2C into seronegative and seropositive would have been better and would show the difference in IL-12-based Th1 cell enrichment.

Our intention for Figure 2A-C was to initially combine all subjects to first characterize what the vaccine is doing to the immune system. Subsequently we focused on transcriptomic rather than flow cytometry responses to assess effects of prior exposure to HSV. Although we appreciate the point about IL-12 and Th1 populations specifically, overall we feel that the transcriptomic responses represent an unbiased approach and have more power based on the greater response observed. Therefore, we have remained focused on the transcriptomic rather than flow data to investigate the effects of prior exposure.

i) Line 300: The authors are correct to note that TLR7 is located on chromosome X. Would it be possible to check within the placebo group comparing males and females if TLR7 is still significantly enriched? This would rule out any vaccine-based effect compared to the sex-based only effect.

See response to essential revisions comment 1.

j) Figure 4 and Lines 334 and 336. Authors state similar pathway enrichments but then conclude that there is a difference in the effect of sex between seropositive and seronegative groups. In Figure 4B, it seems like the biggest difference is seen between seronegative individuals. And yet there are the same genes enriched in.

The contrasts assessed in Figure 4 demonstrate that effects of sex within either seropositive group are statistically different from the effects of sex within seronegative individuals. The text, now lines 336-339, then highlights that the pathways which differed most in both of these comparisons were also those which differed when effect of sex was compared within seronegative subjects alone (Figure 3). These results confirm that the effect of sex on transcriptional response to HSV529 vaccination, which we first observed to be strongest for seronegative subjects in Figure 3, is statistically significant when comparing between naïve and HSV-exposed individuals.

k) Figure 4A. Perhaps some additional explanation could reconcile this contradiction in this paragraph!Line 370: So if I understand correctly, seropositive females lose the high IFN response because of a higher neutralizing titer (as seen in seronegative/positive males?)

Our interpretation is that high early IFN response by HSV naïve women may have inhibited the number of cells this live-attenuated vaccine could infect, and this impaired the titer of the subsequent antibody responses. Please see discussion lines 423-425.

l) Data was not available and is currently embargoed until October 2024!

We will make the data available when the paper is accepted.

Reviewer #2 (Recommendations for the authors):The authors study responses to a non-replicating human HSV vaccine. They find that the vaccine responses differ in males and females, predominantly in those individuals who are HSV seronegative.Strengths.The authors have a relatively large cohort of patients enabling longitudinal studies in seronegative and seropositive male and female individualsThe authors performed a comprehensive study of the immune responses both on the level of neutralizing antibodies and gene activation for long time periodsTheir work identified clear differences between males and females in seronegative individuals: females produce neutralizing ab's faster, but males have delayed but higher levels.The work is able to relate some of the differences to the differences in the Type I IFN response.Weaknesses.Many of the observations reported in the paper have already been described in the literature, such as sex-specific immune responses to viral vaccines.

Sex-specific immune responses to viral vaccines.

We agree sex effects have been reported. However, the combined effects of sex and prior exposure have not. Here we evaluated three different subject groups to dissect the effects of sex when in subjects with: (1) no prior exposure, (2) exposure to prior HSV1, and (3) exposure to prior HSV2. We found significant differences in the effect of sex between these groups, both in innate and adaptive responses.

The possible causal connection between the kinetics of gene activation and the differences in ab responses is discussed but requires further work.

We agree but we feel that further analysis is beyond the scope of this current work. For example, further work will be necessary to determine if the more robust IFN responses observed here in women can be replicated in other naïve responses to vaccination with replication-defective or live attenuated vaccines, and whether such more robust early responses can lead to similar consequences for differential induction of antibody titers. If so, the possibility that a robust early IFN responses in HSV naïve women can inhibit infection or antigen expression by live vaccines would warrant further investigation.

The paper leaves uncertain if this vaccine is going to be epidemiologically/clinically useful (as these are Phase 1 clinical trials).

The prior clinical study report on this vaccine (Dropulic et al. now reference 16) discusses these points. Briefly, the HSV529 vaccine was safe and elicited modest levels of neutralizing antibody responses in HSV-seronegative vaccinees indicating potential as a prophylactic vaccine, although modifications will likely be required to improve its immunogenicity. This paper focuses on the unique opportunity of a cohort with a controlled immune stimulus of vaccination across subjects stratified by both sex and prior exposure, enabling study of how these variables interact to shape an immune response in vivo.

I am uncertain what is the significance of these results for understanding the fundamental biology of the immune responses to vaccination.

This paper emphasizes the interplay between prior exposure to viruses and sex-dependent responses to vaccination. Immune imprinting due to prior exposure is becoming an increasingly important topic affecting the response to many pathogens and vaccines, and sex-specific differences in immune responses to vaccination and infection and disease are becoming increasingly appreciated as fundamental to vaccinology. Our work point to the importance of studying both together as their effects could be dependent on each other.

I think this is a technically very strong paper that reports interesting resultsHowever, I am not sure it reports a breakthrough in our understanding of either the vaccine responses or the sex-linked differences.I think this is important work that might be more appropriate in a human immunology/vaccinology journal.

Please see responses to reviewer 2’s earlier comments above.

Reviewer #3 (Recommendations for the authors):This study describes early antiviral innate responses of a replication-defective herpesvirus vaccine and the subsequent impact on adaptive responses based on the sex of the vaccinee and prior exposure to the virus. As early as one day after the vaccination the differences depended on sex and prior exposure to the virus were noted using blood transcriptomics and cell population profiling. Type I interferon (IFN) signature was found to be the most pronounced in HSV naive women and associated with lower neutralizing antibody titers, suggesting that a strong early antiviral response reduced the uptake of this replication-defective virus vaccine.Strengths:The methodology used by the authors has a wider implication beyond the replication-defective herpes virus vaccine.This is the first detailed study using the systems immunology approach to assess early immune responses and transcriptomic analysis for a replication-defective herpesvirus vaccine.The authors have provided a robust data set to justify the conclusions.Figures are clear and statistical analysis is vigorous.Weaknesses:Overall, this manuscript has minimal weaknesses.The manuscript shows that neutralizing activity is negatively correlated with IFN type 1 signatures in females, it would be of great interest to analyze whether overall gD2 binding or HSV-2 binding antibody levels are also lower in females compared to males or only the antibody with neutralizing activity.If total binding antibody levels are not different, analyzing the Fc receptor-mediated activity such as ADCC would be of great interest.

While we have data on gD2 binding, we focused on neutralizing antibody here as it is thought to be more important as a correlate of protection than simply binding to the soluble gD we have used in our assay. We do not have data on ADCC on most of the subjects in this study. We feel these experiments are beyond the scope of this paper.

A few recommendations are made to improve the manuscript1. Since this study most likely used the same sets of samples that were used in a prior study (Dropulic et al. JID 2019 published by the same group), it may be possible to discuss whether the difference in binding antibody was noted. Alternatively, the data may be further analyzed by sex for discussion purposes.

While we have data on gD2 binding, we focused on neutralizing antibody here as it is thought to be more important as a correlate of protection than simply binding to the soluble gD we have used in our assay.

2. Day 1 following 1st immunization seems to show the most interesting pattern in HSV naïve population. Conceptually, with a replication-defective viral vaccine, the subsequent immunizations in the HSV naive group are more likely to behave similar to that of the virus-exposed group and consequently, immune responses may be indicative of that. It is not clear whether later time point samples were assessed and analyzed based on the naive and prior exposure group.

While we have data on gD2 binding, we focused on neutralizing antibody here as it is thought to be more important as a correlate of protection than simply binding to the soluble gD we have used in our assay.

3. Volcano plot for figure 2B clearly shows the changes in individual genes in samples collected over time. The Y-axis scale variations among panels minimize the appearance of huge differences. If it is difficult to maintain the same scale for all the panels, a comment can be inserted in the figure legend to note the differences in the scale.

We have added to the figure legend in 2B. “Note that the scale of the y axes different on the different days that were analyses.”

4. Previously published phase III trials with subunit vaccines showed higher efficacy in the HSV naïve women. It is to be noted that higher efficacy in females was not assessed for correlation with neutralizing activity in the prior study. One of the most interesting findings presented here is that a replication-defective vaccine had a smaller boost in neutralizing activity in females following 2nd immunization compared to men. It may be interesting to discuss the possibility of variations in immunological outcome based on the sex with a replication-defective viral vaccine compared to an adjuvanted subunit vaccine.

Please see response to essential revision comment 3 above.